# Electroencephalographic Biomarkers for Neuropsychiatric Diseases: The State of the Art

**DOI:** 10.3390/bioengineering12030295

**Published:** 2025-03-14

**Authors:** Nayeli Huidobro, Roberto Meza-Andrade, Ignacio Méndez-Balbuena, Carlos Trenado, Maribel Tello Bello, Eduardo Tepichin Rodríguez

**Affiliations:** 1School of Biological Sciences, Universidad Popular Autónoma del Estado de Puebla, Puebla 72000, Mexico; 2Departamento de Ciencias de la Salud, Universidad de las Américas Puebla, Puebla 72000, Mexico; roberto.meza@udlap.mx; 3Facultad de Psicología, Benemérita Universidad Autónoma de Puebla, Puebla 72000, Mexico; 4Institute of Clinical Neuroscience and Medical Psychology, Medical Faculty, Heinrich Heine University, 40225 Duesseldorf, Germany; carlos.trenadoc@gmail.com; 5Escuela de Ingeniería y Actuaría, Universidad Anáhuac, Puebla 72000, Mexico; maribel.tello@anahuac.mx; 6Optics Department, Instituto Nacional de Astrofísica, Óptica y Electrónica, Puebla 72000, Mexico; tepichin@inaoep.mx

**Keywords:** qEEG, biomarkers, machine learning, LORETA, depression, Alzheimer’s disease, migraine, TBI, schizophrenia, COVID-19

## Abstract

Because of their nature, biomarkers for neuropsychiatric diseases were out of the reach of medical diagnostic technology until the past few decades. In recent years, the confluence of greater, affordable computer power with the need for more efficient diagnoses and treatments has increased interest in and the possibility of their discovery. This review will focus on the progress made over the past ten years regarding the search for electroencephalographic biomarkers for neuropsychiatric diseases. This includes algorithms and methods of analysis, machine learning, and quantitative electroencephalography as applied to neurodegenerative and neurodevelopmental diseases as well as traumatic brain injury and COVID-19. Our findings suggest that there is a need for consensus among quantitative electroencephalography researchers on the classification of biomarkers that most suit this field; that there is a slight disconnection between the development of increasingly sophisticated methods of analysis and what they will actually be of use for in the clinical setting; and finally, that diagnostic biomarkers are the most favored type in the field with a few caveats. The main goal of this state-of-the-art review is to provide the reader with a general panorama of the state of the art in this field.

## 1. Introduction

Adequate treatment of disease cannot rely solely on the patient’s subjective description of their symptoms. Thus, there is a need for reliable and measurable correlates of the signs of disease and its progress. These are known as biomarkers: characteristics that can be measured objectively and evaluated as indicators of normal biological processes, pathogenic responses, or pharmacologic responses to a therapeutic intervention [1]. Biomarkers have been used with much success for a long time in clinical practice, but it was the development of specialized technologies that allowed for the development of increasingly specific and sophisticated biomarkers.

Consequently, nowadays, biomarkers are not only tools for corroborating a diagnosis and for monitoring the progression of disease. Instead, some have turned into the sole conclusive tool for the detection or identification of illness. Complex syndromes of uncertain etiology can now be prevented and treated thanks to the use of biomarkers; however, there is an ever-present need for more specificity in treatment and more consensus among health professionals. Those that pose significant challenges for clinicians because of their incidence or complexity include metabolic, cardiac, immune, and hormonal diseases. Yet, those that have garnered the most attention in the past three decades are neuropsychiatric diseases.

In short, neuropsychiatric diseases are those in which a combination of neurologic and psychiatric symptoms is present, for instance, Parkinson’s disease, Alzheimer’s disease, and mood disorders, among others. Neurologic symptoms are observable (albeit some only postmortem) and are related to changes in neural tissue that may cause, for instance, changes in perception, movement, awareness, or wakefulness. On the other hand, psychiatric symptoms affect aspects of the patient’s cognition and executive functions, including emotional regulation, memory, etc., which can hinder the patient’s ability to describe and explain their experience. Not all neurologic disorders correlate with psychiatric symptoms, and vice versa. Thus, neuropsychiatric diseases can also be defined by the approach to their study: the consideration of the subjective experience of mental states in the first-person perspective over the sole objective scientific observation of neuronal states [2]. Therefore, their diagnosis and treatment are complicated due to the following: symptoms overlap and can vary significantly among patients and even different populations; moreover, they are subject to the clinician’s interpretation. These facts are attested by the controversial nature and constant flux of psychiatric disease classification. All of the above lead to difficult and often frustrating treatment strategies, which affect the patient’s quality of life. It is evident for both patients and clinicians that there is a sore need for better diagnostic tools, namely biomarkers for neuropsychiatric diseases. A useful strategy for their development has been to build on existing instruments. Exploiting electroencephalography has been particularly fruitful. Some psychiatric diseases share features that make them hard to distinguish. This translates to treatment overlaps and widely varying degrees of success. Biomarkers could make diagnosis and treatment more precise, therefore increasing rates of success and the patient’s wellbeing. A shift in the conception of the etiology of some neuropsychiatric disorders (as in the serotonergic hypothesis of depression, the vascular nature of migraine, etc.) means that electroencephalographic activity is seen as a putatively more reliable tool with which to replace less precise methods or subjective practices.

Diseases affecting the brain are understood to affect circuits, and the shifts in their function are mostly reflected in cortical activity. Consequently, biomarkers can be obtained through electrophysiological tools such as electroencephalography (EEG) and magnetoencephalography (MEG) (for review, see [3]). However, while some diseases (mostly neurologic) have characteristic qualitative EEG patterns (those that can be found through visual inspection), this is not usually the case for psychiatric disorders. Moreover, psychiatric diseases generally do not have an anatomical correlate that can be detected in imaging studies. Thus, finding biomarkers in EEG is mostly a matter of refinement: sophisticated numerical and statistical methods must be applied to the signal in order to find the hidden features that might give away specific illnesses. This approach is known as quantitative EEG (qEEG), and it poses particularly interesting problems for engineers, who have poured significant effort into the endeavor. Recent developments of this approach and its caveats are the subject of the present review. A brief overview of EEG terminology is offered in the next section of this paper.

Another important incentive for the progress of biomarkers for neuropsychiatric diseases is the existence of transitional stages: periods of apparent stasis that are, effectively, steps in the progression of a disorder. Their identification may facilitate the adoption of preventive measures. In traumatic brain injury (TBI), for instance, electrophysiological biomarkers could be a valuable aid in monitoring the progression of the lesion and in tailoring neuropsychological and pharmacological treatments that may slow down the modification of cognitive faculties. Different stages in the development of cognitive deficit are also a target of this field.

Outside of very specialized circles in clinical research, the term biomarker usually has a broad, oftentimes implicit definition. It includes a diversity of biological and clinical phenomena and even extends to any interaction between a biological system and a potential hazard [4]. This situation varies depending on what disease is being discussed and from what point of view. It is no different for neuropsychiatric diseases: additional effort is needed to persuade all authors in the field to explicitly define what they consider a biomarker and how theirs could be classified. The need for neuropsychiatric biomarkers is clear; how each new reported candidate fits into the greater scheme and how it may benefit clinicians and patients is not always so straightforward. On the other hand, it should be acknowledged that most broad definitions of biomarkers consider them biochemical entities and/or tools for drug development, none of which fits the definition of electroencephalographic biomarkers. At least two classifications have been published specifically for biomarkers for neuropsychiatric diseases. Weickert et al. [5] proposed three classes, while Davis et al. [6] provided six potential categories (Table 1). While Weickert et al.’s approach is more succinct, Davis et al.’s more thorough proposal is a better fit for the occasional ambiguity that some published papers show regarding what their proposed biomarker(s) is intended for. Table 2 summarizes recent representative examples of biomarkers from the literature; in each case, we identify the type of biomarker proposed according to Davis et al.’s classification.

Regardless of the classification of EEG biomarkers from a clinical point of view, in technical terms pertaining to signal analysis, they are often categorized into three primary domains: frequency-domain, time-domain, and connectivity-based metrics. The algorithms designed for qEEG biomarkers have the properties of specificity, sensibility, and efficacy [26]. Each algorithm or set of algorithms will perform better in one of these criteria or a combination thereof. This explains why the same measure of connectivity (alpha power, for instance) has been tested by several authors on patients with the same disease. Currently, this is a feature of the qEEG biomarker field. The reader should be cautious when consulting the literature and try to identify the actual novelty or distinguishing feature of each paper since the amount of published works, algorithms, and abbreviations can be overwhelming. This is especially true for readers without an engineering or mathematical background.

The present brief state-of-the-art review is intended to give the reader a general panorama of the field by summarizing representative works from the past decade (2014 to 2024). The diseases covered were chosen because of their significance both within and outside of the field, namely cognitive impairment, including mild cognitive impairment, dementias, and attention-deficit hyperactivity disorder (ADHD); TBI; depression; migraine; and epilepsy. Cited works were sourced from PubMed using combinations of keywords like “qEEG”, “biomarkers”, and “electroencephalography/electroencephalographic” and the names of the diseases treated in the review. A profound methodological analysis is beyond the scope of this type of review. The target audience is readers new to the field, either from engineering or the life sciences.

## 2. A Brief Summary of Relevant Terminology

In this section, we offer the reader a few definitions relevant to the following discussions. Those familiar with EEG terminology may skip this section. Further details can be found in the literature recommended within the text.

*Electroencephalography* (EEG) is the main technique used to record (graphia) the electrical activity (electro) of the brain (enképhalos). Electrodes are placed on the scalp through which electrical currents originated in the brain are sent to special devices that amplify, filter, and then display and/or store them. It is not invasive, comparatively economical, and very well characterized, which is why it is so important in research and clinical practice despite its age. An additional advantage is its excellent temporal resolution in the order of milliseconds using specialized equipment, which is not attainable with imaging techniques such as functional magnetic resonance (fMRI). It is important to emphasize, for the novice, that what is recorded does not includeaction potentials, which are too weak to be detected by the electrodes, but *post-synaptic potentials* that occur in the neocortex [27]. Neurons are organized in complex circuits; collectively, the aforementioned post-synaptic potentials generated in the circuit produce *oscillatory activity*. Thus, brain activity is recorded as waves, which in turn represent the spatial and temporal summation of smaller, localized electrical events. These waves are organized in five main *frequency bands* (several subsets have been described but are beyond the scope of this summary): delta (0.5–4 Hz), theta (4–7 Hz), alpha (8–13 Hz), beta (13–30 Hz), and gamma (>30 Hz). Each of these has been correlated to a different (ever increasing) group of events, such as quiet wakefulness, sleep, focused thinking, visual stimuli, and many more. Being waves, they can be separated into their *power and time-domain components* [28]. Circuits activate or inhibit each other and otherwise cooperate to form the whole variety of brain activity. Communication among neural circuits must be cohesive to yield useful results; in other words, it must have coherence, a measurable property that varies in distinct ways in different brain disorders. This and other similar measures are collectively known as *connectivity-based metrics*.

For the most part, an EEG analysis for clinical purposes is *qualitative*. Traits such as distinct wave patterns, frequency, amplitude, and the very presence or absence thereof, along with the patient’s characteristics (such as age, sex, illnesses in other parts of the organism, etc.) are observable by the naked eye and give clinicians enough information to identify and treat a large group of neuropsychiatric conditions. EEG manuals contain information on electrode placement configurations and methods of analysis (EEG primer). The clinician or researcher can obtain information from a patient’s brain by administering different types of stimuli known as *activation procedures* [27]. These may include sleep deprivation, bouts of hyperventilation, photostimulation with strobe lights of different frequencies, etc. Importantly, the lack of stimuli is also an important diagnostic tool; this is known as the *resting-state* EEG. The recording is performed on the awake, quiet patient, with no other task or stimulus than the order to remain with their eyes open, then closed, and finally open again. This is reported in the literature in slightly different ways, such as the *eyes open, eyes closed task*, the eyes closed (or eyes open) condition, etc., always referring to the same technique. The importance of this method lies in the fact that distinct states of brain activity will activate or block waves, which in turn suggests what phenomena (or aspect thereof) these might be involved in. Methods of EEG analysis must consider any task given to the patient; specific combinations of tasks may be designed to provoke known states of activation, which may vary among neuropsychiatric disorders [28]. Investigators should always bear in mind the fact that the normal EEG signal differs among age groups because this can be an important confounding variable. Significant training is necessary for the practical use of EEG in a clinical setting because there can be important overlaps in each of the variables mentioned above among subjects and disorders. This type of analysis is very useful but, inevitably, far from perfect and prone to errors.

Several decades ago, investigators [29] reasoned that there should be complexities hidden among the observable features in EEG that could yield more precise (and more intricate) data. Complex mathematical methods are needed to bring these to the surface, such as *fast Fourier transform* (one of the first techniques used), *wavelets*, several types of *filters*, etc. Such algorithms and computations, in turn, require the use of equally complicated technology which, in the past 25 years or so, has become increasingly accessible and therefore applicable beyond the walls of research laboratories. This modality of EEG manipulation and analysis is called *quantitative EEG* (qEEG). Researchers and clinicians alike trust that the use of qEEG will enhance diagnostic precision and objectivity. It has already made it possible to study conditions for which qualitative analysis is of little use either because it shows apparently no distinguishing features or because these become confounded with those of other illnesses. Controversies around misdiagnoses of affective disorders, for example, arise due in part to these very reasons. Work on the disorders discussed in the present review, among others, has greatly benefited from qEEG. This fact alone justifies the rapid growth and complexity of this recent field of neuroscience. A relevant feature of brain activity that has been used in research for decades is the emergence of low-amplitude, rapid, circumscribed responses to specific sensory stimuli. These are known as *event-related potentials (ERPs)* [30]. Using ERPs requires the recording and averaging of several responses in order to obtain a useful signal. Current qEEG research makes extensive use of ERPs as a tool for finding very specific characteristics in diverse neuropsychiatric diseases.

One of the issues of EEG, due to inherent features of both the brain and of the technique itself, is dubbed the *inverse problem* (see [31] for a thorough review). The name means, briefly, that while the result of brain activity can be detected, it is impossible to pinpoint its exact origin. This implies that the electrical activity of the neocortex is not generated there exclusively; actually, most of it arises in the thalamus, other subcortical nuclei, and parts of the cortex which can be far away from the site in which the activity is being recorded. The main method that has been developed to elucidate the source of activity is called *LORETA* or Low-Resolution Electromagnetic Tomography [31]. LORETA works on the EEG recording itself and produces maps of putative sources (such as the thalamus, insula, cerebellum, cingulate cortex, basal ganglia, etc.) of the signals. qEEG incorporates the use of LORETA to either hypothesize or corroborate (for practical purposes) the origin of pathological EEG patterns.

Lastly, readers developing in areas outside of life sciences should not forget that what is known about the neurophysiological correlates of EEG activity (that is, which neurons, localized where generate what, and how) comes from three main sources, namely decades of clinical observations, even more decades of lesion and pharmacological experiments on animal models, and still more decades of the analysis of lesions in human subjects, mostly war veterans. Mathematical models of brain activity owe their bases to these experiments and analyses, whose replacement with technological means is simply impossible.

## 3. Cognitive Impairment

Cognition is arguably the most important brain function for humans. It is the result of binding thought, memories, and perception which allows for the acquisition of knowledge and its comparison with past experience. Cognitive impairment is an umbrella term that refers to a gradual decline in a person’s ability to learn, remember facts, and make decisions, among other mental processes. Features of cognitive impairment include trouble concentrating, understanding, solving problems, etc. It can be considered a spectrum, whose initial stage is usually related to age and could be considered by most people as mere forgetfulness. The other end of the spectrum is known as dementia: “a syndrome of cognitive impairment or cognitive decline that affects independent living [32]”. Cognitive impairment may be mild or severe with regard to its impact on a person’s quality of life. In recent years, a person’s worry that their cognitive abilities may be in decline, which often leads to seeking medical attention, has been termed Subjective Cognitive Impairment (SCI). This stage is being increasingly discussed as a possible very early stage of dementia because, while it cannot be objectively tested for (at the time of this publication), there is evidence that a significant portion of adults who report it eventually enter the next stages of cognitive decline [33,34]. The next step, mild cognitive impairment (MCI), is often considered a transitional stage between the expected cognitive decline of getting older and the more serious cognitive decline of dementia [35,36]. People with MCI may be aware that their mental ability has changed, but such changes usually do not impact their daily life activities. Importantly, MCI increases the risk of developing dementia caused by Alzheimer’s disease or other neurological conditions. In contrast with SCI, MCI can be tested for. The development of both stages is considered an important step towards their consolidation as diagnostic entities as well as in the management of patients who report them.

## 4. Resting-State EEG

Resting-state EEG has been used by a significant number of authors for the development and refinement of biomarkers. Especially in the case of MCI and dementias, the resting state is arguably the default EEG recording of choice. Dementias and cognitive impairment are the result of brain degeneration and, therefore, the disappearance of neuroanatomical, functional, and effective connectivity. Simple measures obtained during the resting state and the eyes closed to eyes open transition are indicative of the activation or inhibition of whole cortical and thalamo-cortical networks. Therefore, the resting state and the eyes open, eyes closed task are considered enough to obtain information on brain connectivity. While refinement of this information is ongoing, these facts are sufficient for the pursual of increasingly more precise and sensitive algorithms for qEEG analysis applied to dementias and cognitive impairment. In the case of AD, dementia with Lewy bodies, and Parkinson’s disease dementia (PDD), measures of spectral power have been found to enable the discrimination of patients with different types of dementia, ill patients from healthy controls, and patients with distinct subtypes of AD. Other authors have used peak frequency as a purported indicator of the status of brain connectivity; recently, Cao et al. [11] reported a new, more precise method for obtaining a reliable measure of peak frequency, with better results than those obtained with methods like wavelet analysis or short-time Fourier transform.

A number of reports found increased alpha and decreased delta power spectrums, as compared to healthy controls, as qEEG features of MCI [7]. Previous EEG/MEG metanalysis studies assessing functional connectivity changes at rest along the healthy–pathological aging continuum reported an alpha synchrony decrease in MCI compared to healthy subjects, specifically between temporal–parietal and frontal–parietal areas [37]. It was also reported that alpha rhythms may be more affected by disease variants such as in Alzheimer’s disease (AD) in relation to early vs. later onset.

Recent EEG studies suggested that cortical activity characterized by microstates and connectivity may be able to differentiate the neural signature of subjective cognitive decline and MCI conditions. It was revealed that people with MCI showed less complex microstate sequences than people with cognitive decline and healthy controls. The EEG spectral content, network connectivity, and the spatial arrangement of microstates were similar in the three groups. Thus, comparing properties of microstates might provide insight into the progression of dementia in relation to degradation of cortical organization [9].

Other MEG graph theory analyses were directed to evaluate the characteristics of the resting-state networks corresponding to patients suffering from schizophrenia, which is characterized by psychotic symptoms and cognitive impairment. In particular, it was reported that associated patient graphs disintegrated mainly in the beta band, thus providing deeper insights into the pathophysiology of the disease and their corresponding symptoms as a disconnection syndrome [38].

An EEG-based biomarker to characterize cognitive functions in Parkinson’s disease from a few minutes of resting-state EEG was proposed on the basis of multiple changes in spectral rhythms [39]. In fact, previous studies showed that interventions such as cognitive training and physical activity can have effects on executive functions and attention with a corresponding modulation of resting-state EEG (theta and alpha power) in patients with PD with mild cognitive impairment [13]. Babiloni et al. [14] observed a marked reduction in alpha rhythm desynchronization in response to visual stimuli compared to healthy older adults and patients with ADD; in other words, desynchronization occurred less in patients with PD dementia than in the other groups. The authors propose this lack of activity as a comparative factor in patients with PDD who undergo neuromodulatory stimulation. This follows from a previous study by Schumacher et al. [12] in which alpha power, upon opening the eyes, was found to be impaired in patients with different types of dementia: in descending order, alpha reactivity was reduced more in patients with PDD, followed by patients with dementia with Lewy bodies (DLB) and patients with AD. Taken together, these studies suggest that the proportion in which alpha reactivity upon opening the eyes is affected, which may facilitate the distinction of patients with different types of dementia. Indeed, Gimenez-Aparisi et al. [15] remarked on the importance of assessing EEG features during specific stages of the eyes open, eyes closed task. The authors specifically tested the eyes closed and the eyes open conditions as well as reactivity to eyes opening. Their results (summarized in Table 2) suggest that extracting a set of features, taking advantage of the relative simplicity of the eyes closed, eyes open task, provides better chances of enabling clinicians to distinguish abnormal neural activity in patients with PD from healthy subjects in order to anticipate the onset of PDD.

An assessment of the relationship between resting-state EEG signatures (power) and cerebrospinal fluid markers (CSF) in an MCI population revealed a negative correlation of Aβ-42 load with central posterior theta power and negative correlation of t-tau with widespread alpha power within the male subsample and a significant negative correlation between t-tau and widespread beta power in the female subgroup [40]. Recently, the data-driven retrieval of population-level EEG features has been proposed as a viable approach to distinguish patients with AD and healthy controls and may also be useful to help characterize MCI [41]. Similarly, Nencha et al. [8] found that qEEG is capable of discriminating between tau- or amyloid-driven AD neurodegeneration. Specifically, they found an increase in gamma oscillations over the frontal and parietal regions during the resting state. The authors did not specify whether this increased power occurs in a particular stage of the eyes open-to-eyes closed transition. In this case, the conclusion is that the presence of tau depositions forces neurons to maintain their normal network function through increased activity.

A concerning effect of the COVID-19 pandemic is the emergence of signs of cognitive decline among affected populations; this has been demonstrated even among individuals with mild cases of the disease [42]. Features among survivors with complaints of Subjective Cognitive Decline include increased central and parietal beta/theta ratios as well as significantly lower frontal, central, and parietal coherence [17]. Importantly, Sun et al. [16] found that issues related to cognitive performance and memory appear in infected patients regardless of age. The authors reported a marked rise in low-complexity, synchronized neural activity within low-frequency EEG bands as well as effects on cognitive functions and brain connectivity, consistent with cognitive decline and EEG disarray similar to those found in ADHD and MCI. Remarkably, younger subjects (age < 10) exhibit milder effects, while young adults (age 20–27) were shown to be a particularly vulnerable population.

## 5. Task-Oriented qEEG for Cognitive and Motor Processing

EEG measures of cognitive processing have been pursued in both healthy individuals and subjects with cognitive impairment. Broitman et al. [43] found that healthy older adults have a lower capacity for memory encoding when compared to younger subjects, which reflects in alpha activity, while both alpha and gamma evidenced differences in task demand. In a separate study, wearable EEG recordings were utilized to assess neurophysiological signatures of MCI during a working memory task in healthy older adults. The authors reported that individuals with a lower memory retrieval accuracy showed significantly increased alpha and beta oscillations over the right parietal site. Faster working memory retrieval was significantly correlated with increased delta and theta band powers over the left parietal sites. Increased coherence between the left parietal site and the right frontal area is correlated with a faster speed in memory retrieval. The frontal and parietal dynamics of resting EEG is associated with the “accuracy and speed trade-off” during working memory [44]. Thus, biomarkers of the state of brain connectivity and reactivity in healthy older adults include differences in alpha desynchronization in the resting state [45] as well as features of the delta and theta bands during cognitive tasks.

Event-related potentials (ERPs) have been studied as biomarkers for MCI. Combined with power spectrum measures, ERPs, as evidence of processing speed and accuracy, can denote features of MCI such as motor imagery deficits [46], emotional working memory tasks [47], short-term memory deficits [48], and executive function impairment [49]. Response delays also help distinguish individuals with MCI-AD from healthy older adults [50], as well as ERP features like the latency of P200, P300, and N200 [51]. A measure consisting of resting-state qEEG and two ERP-related tasks has been proposed as a tool with diagnostic, state, and treatment assessment capabilities for the assessment of early cognitive decline in AD [10]. This type of development could benefit clinicians since the biomarkers included in the measure complement each other. However, whether similar tools can be developed for other neuropsychiatric disorders will depend on the amount and quality of work made for each particular case.

With regard to attention-deficit/hyperactivity disorder (ADHD), which leads to decline in academic performance, interpersonal relationships, and development, previous quantitative EEG (QEEG) studies searching for biomarkers of ADHD have considered the theta/beta ratio (TBR), phase-amplitude coupling including theta phase-gamma amplitude coupling [52], global weighted coherence, and delta/theta power [53,54]. According to a systematic review by Michelini et al. [55], specific increases and decreases in frequency band power observed in ERPs related to specific cognitive and memory tasks are consistent across studies. However, the authors identified the influence on small populations, as is the case for other neuropsychiatric diseases. Moreover, it is necessary to focus research efforts on obtaining clinical profiles (state/acuity biomarkers) instead of simply identifying the diagnostic status.

## 6. Traumatic Brain Injury

Electrophysiological measures offer critical real-time insights into brain function and dysfunction following TBI, a major global public health issue. TBI frequently leads to long-term cognitive, emotional, and physical impairments and remains a leading cause of morbidity and persistent neurological deficits worldwide.

EEG has proven especially effective in identifying abnormalities associated with mild TBI. Research demonstrates that EEG can uncover disruptions in neural connectivity and changes in brain wave patterns that are hallmark features of mild TBI. Furthermore, advances in computational methods, such as machine learning algorithms applied to EEG data, have significantly improved the accuracy of mild TBI detection. These innovations pave the way for rapid, non-invasive diagnostic approaches, enhancing the timeliness and precision of TBI assessment [56]. This integration of electrophysiological tools into TBI care underscores their importance in advancing diagnostic and therapeutic strategies, enabling better outcomes for individuals affected by this condition.

Frequency-domain biomarkers, such as changes in spectral power, have been extensively studied in TBI research. For example, increased delta (0.5–4 Hz) and theta (4–8 Hz) power is frequently observed following TBI, reflecting deafferentation and structural damage. These changes have been shown to correlate with the severity of the injury [57]. Studies have linked heightened delta activity to poorer cognitive outcomes and reduced recovery potential, including executive function complaints, lower premorbid IQ, poorer cognitive performance, and higher psychological distress symptoms [58]. Additionally, EEG assessments post-TBI frequently show slowing of the posterior dominant rhythm and increased diffuse theta slowing, which may revert to normal levels within hours or may clear more slowly over many weeks [59]. Additionally, studies have found that post-traumatic amnesia (PTA) and loss of consciousness (LOC), along with poorer cognitive performance, are associated with reduced power in beta frequencies, while executive function complaints and higher psychological distress are linked to increased delta power [60].

Reduced alpha and beta activity (α: 8–12 Hz, β: 13–30 Hz) is frequently observed in TBI, correlating with impaired cognitive function and reduced neural synchronization. In particular, alpha suppression has been associated with attention deficits and slowed processing speed. Frontal Alpha Asymmetry (FAA), where there is a difference in alpha power between the left and right frontal hemispheres, has gained attention as a biomarker for mood and emotional regulation following TBI. Abnormal FAA patterns have been associated with depression and anxiety, which are prevalent in patients with TBI [61]. In Theta/Beta Ratio (TBR), an elevated TBR is a marker of frontal lobe dysfunction, frequently used in mild TBI (mTBI) assessments and neurofeedback [60]. These findings underscore the significance of power spectral density as biomarkers for assessing the extent of brain injury and predicting cognitive recovery trajectories in patients with TBI [61]. By capturing these patterns, EEG biomarkers provide a robust framework for understanding the neural disruptions caused by TBI and offer critical insights for guiding therapeutic interventions.

Regarding time-domain biomarkers, Shifts in Slow Cortical Potentials (SCPs) are associated with disruptions in cortical excitability and self-regulation, providing insights into recovery mechanisms in patients with TBI. Changes in somatosensory, auditory, and visual Evoked Potentials (EPs) are indicative of sensory and cognitive processing deficits in patients with TBI [62]. In addition, event-related potentials (ERPs) such as P300, N200, and N400 components are frequently used to assess cognitive deficits post-TBI. Delayed or reduced P300 amplitude and delayed latency are common in patients with TBI, indicating impaired attentional processing and working memory. N200 alterations have been linked to impaired cognitive control and conflict monitoring [63]. These ERP alterations are particularly useful in detecting mild TBI, which often lacks structural abnormalities on imaging.

Functional Connectivity Biomarkers are measures of EEG coherence, phase-locking value in EEG, and phase synchronization between brain regions. These measures reveal disrupted inter-hemispheric communication in TBI and therefore provide insights into inter-regional brain communication [64]. These alterations are prominent in default mode and frontoparietal networks. Coherence in delta and theta bands has been proposed as a potential biomarker for distinguishing mild from severe TBI [65]; weighted coherence has also been put forward as a possible measure of TBI progression [19].

Finally, EEG microstates are brief, quasi-stable topographic patterns in EEG that reflect functional brain networks. Altered microstate dynamics, such as reduced durations or transitions between specific states, have been linked to cognitive and emotional deficits in patients with TBI [66].

### 6.1. Advances in EEG Analysis for TBI Biomarkers

Recent advancements in computational neuroscience have enhanced the utility of EEG in TBI research with techniques such as Graph Theoretical Metrics in which a network analysis of EEG data showed reduced global efficiency and increased modularity in patients with TBI, indicative of impaired network integration and compensatory reorganization. The convergence of electrophysiological data with machine learning techniques has led to significant advancements in TBI diagnostics. By analyzing complex EEG patterns, machine learning models can classify mTBI with high accuracy. In machine learning approaches, algorithms trained on EEG features can classify TBI severity, predicting outcomes and identifying subtypes of injury with high accuracy, facilitating early intervention and personalized treatment strategies [67]. High-density EEG analysis can improve the spatial resolution facilitating the detection of focal abnormalities and enhancing the connectivity analysis.

### 6.2. Clinical Applications

EEG biomarkers have been utilized to monitor recovery trajectories and predict outcomes in patients with TBI. For instance, delta power normalization during rehabilitation is indicative of recovery, whereas persistent abnormalities suggest poor prognosis [68]. Portable EEG devices have further enabled real-time monitoring of brain activity, broadening the accessibility of EEG biomarkers in clinical settings [69]. In acute assessment, rapid EEG screening can assist in detecting TBI severity and identifying patients at risk of secondary complications. Already, a few commercial devices are available that promise to provide professionals at emergency rooms and other points of care with diagnostic information for patients with TBI [70]. Devices of this sort usually hold their methods of analysis under secrecy. Unfortunately, whether they deliver on their promises reliably or not is tested directly on patients by unknowing physicians.

Finally, qEEG technologies have also been applied to rehabilitation monitoring in patients with TBI. Particularly, brain–computer interfaces (BCIs) allow patients to track their own progress over time through biofeedback and other techniques. For instance, BCIs have been effectively employed in motor rehabilitation for individuals with neurological disorders, particularly stroke, demonstrating improvements in motor function [67,71,72].

Transcranial magnetic stimulation-associated measures of cortical function and plasticity, such as short-latency afferent inhibition, short-interval intracortical inhibition, and the cortical silent period, may add useful information in most cases of secondary dementia, especially in combination with suggestive clinical features and other diagnostic tests [73]. Recent AD studies making use of emerging stimulation techniques, namely transcranial pulse stimulation, emphasize electrophysiological markers such as spectral power (frontal and occipital), coherence (frontal, occipital, and temporal), Tsallis entropy (temporal and frontal), and cross-frequency coupling (parietal–frontal, parietal–temporal, and frontal–temporal) [74].

## 7. Depression

Depression is one of the neuropsychiatric disorders that are better known by the general public, both because of its incidence and its effects on patients’ quality of life. Because of the complex process of accurate diagnosis and treatment selection (usually by trial and error), quantitative biomarkers are sorely needed. Efforts during the past decade have found that alpha and theta band rhythms provide clinically useful information about this neuropsychiatric disease. This includes disease progression, response to treatment, treatment selection [21], and features distinguishing it from other mood disorders (Sun et al. [22] used phase lag index instead of coherence as a more robust measure of connectivity). In recent years, the gamma rhythm has been investigated for the same purpose as well (see Fitzgerald and Watson, 2018, for review [75]). Van der Vinne et al. [21] tested the possibility of using qEEG features in patients with depression to aid in the selection of suitable pharmacological treatment. Frontal alpha asymmetry and several types of abnormal activity determined the choice between venlafaxine, escitalopram, or sertraline. The authors reported good outcomes from the points of view of the physician and patient satisfaction. Regarding the actual effectiveness of the antidepressant choice for each patient, the size of the effect was medium. Regardless, the authors argue that their success contradicts the conclusions of a meta-analysis by Widge et al. [76], which state that qEEG is not recommended for guidance in the selection of psychiatric treatment because of several serious methodological problems and publication bias found in the reviewed studies.

The field of qEEG biomarkers in general suffers from a lack of methodological consensus and a lack of adequate validation, which lead to difficulties in the interpretation of different studies and reduce the possibility of clinical application. In this context, a lack of methodological consensus refers to the fact that a given feature is analyzed by several different groups, each with a distinct set of analysis methods. This work yields data about the performance of these algorithms, but the relationship of the proposed biomarker with its original intent is diluted. Smith et al. [20] addressed this by comparing the validity of two candidate resting-state biomarkers for depression, namely rostral anterior cingulate cortex theta (extracted with LORETA) and posterior alpha identified with scalp current source density, in healthy subjects. The authors were able to identify inconsistencies in the first biomarker, recommending that it be used with analyses other than those reported in the literature thus far. Papers like this emerge at a slow rate, while those that test the latest algorithm for its own sake abound.

## 8. Migraine

The importance of migraine among other neuropsychiatric diseases stems from two reasons: first, it is classified as the sixth most disabling disease worldwide [77]. Second, research on the neurophysiological causes of migraine has led to new insights on the influence of structures like the medulla, the thalamus, and the trigeminocervical complex, among others, on cortical function. Current prevailing hypotheses go well beyond the putative origin of migraine as a vascular disorder. For instance, O’Hare et al. [24] used qEEG to investigate features of brains with migraine and found differences in performance during visual tasks due to lower alpha band power. This is indicative of a characteristic diminished cortical excitability that has been observed previously. While the authors argued that certain tasks could compensate for this feature, they would not be powerful enough to cause neurophysiological changes and would therefore have little use as a treatment for this disorder. Other researchers have focused on distinguishing neurophysiological features of chronic migraine. Using intranasal trigeminal stimulation, Haehner et al. [78] found that there are differences in the processing of nociceptive stimuli from the trigeminal nerve in patients with migraine. These involve the temporal lobe, the cerebellum, and the precuneus, all of which are related with some aspect of the sensory or emotional processing of noxious stimuli. Measures of brain connectivity are particularly relevant to migraine research; in particular, microstates have garnered considerable attention, in combination with fMRI, neuropsychological test batteries, etc. With these tools, Zhou et al. [79] found that the connectivity of resting-state networks in patients with migraine correlate with their degree of migraine-related disability.

Lastly, EEG recordings from patients with migraine or epilepsy were used to train three machine learning techniques: deep LSTM, a reservoir spiking neural network, and NeuCube, a brain-inspired spiking neural network. The objective was to distinguish between both disorders through their qEEG features. The authors obtained promising results regarding the performance of the machine learning algorithms used [25]. Interestingly, it was shown that this task could be achieved even with a small dataset obtained from only 21 subjects. As good as these machine learning results are, it should be pointed out that from the point of view of neurophysiology, the importance of this type of research is dubious. At the very least, authors should be careful not to oversell the potential clinical value of their methods since this may undermine their credibility.

## 9. Epilepsy

Epilepsy is a neuropsychiatric condition characterized by exacerbated neural activity that may be localized in different sites of the neocortex. Its incidence is estimated at around 67.77 per 100,000 persons [80], with generalized seizures and those of unknown etiology being the most prevalent. Epilepsy affects patients’ quality of life considerably; when untreated, it can have important neurological consequences. At the same time, seizures are a rather common neurological symptom which is not always correlated with the risk of developing active epilepsy. For these and other reasons, a considerable amount of effort is poured into the diagnosis and follow-up of this disease. Therefore, biomarkers for epilepsy could be of considerable value for the clinician. Recent efforts include the report by Klotz et al. [23], who found that high-frequency oscillations of the scalp can be reliable predictors of epilepsy after a first unprovoked seizure in childhood. While very-high-frequency oscillations have been shown to be even more efficient biomarkers, these are recorded during surgery using deep brain electrodes [81]. Both of these measures are EEG oscillatory activity with a frequency >100Hz, which is of apparently little use under qualitative analysis.

## 10. Discussion

Woodcock et al. [82] argued that new biomarkers are usually adopted without a formal demonstration of clinical utility, after medical specialists (or associated entities, in this case, including tech entrepreneurs) use what is available in scientific publications, without regard to the quality of the experimental design, statistical analysis, or basic premise. While several clinical studies with greater cohorts exist that explore the feasibility of biomarkers for specific neuropsychiatric diseases, most biomarker candidates have not reached this state. If the fact that most algorithms used are opaque for clinicians is considered, this means that biomarkers could be adopted “blindly”, especially when adopted indirectly, through the use of new technologies with proprietary, even more cryptic, methods of analysis. Of course, one would not expect clinicians to develop their own algorithms or even thoroughly grasp those that are published in the same way that people are not expected to build their own cars and trust that they will work as intended. As a consequence, and for the benefit of both clinicians and patients, those who publish new biomarker candidates should be as transparent as possible. On the other hand, clinicians should be aware that, like any other diagnostic technique developed under ideal laboratory conditions, qEEG biomarkers are investigated with limited populations and idealized sets of stimuli, which include patients whose disease is stable in order to avoid confounders. This means that the application of qEEG biomarkers cannot replace the trained eye of the physician, electrophysiologist, or neurophysiologist.

Methodological heterogeneity in the field of neuropsychiatric biomarkers makes it difficult to achieve consensus, cross-study comparison, replicability, and the redirection of research efforts towards clinical application [83]. However, not all diseases will benefit from the same methodological approach. As stated above, one recurrent method is to record resting-state EEG. Several examples exist of this approach for different diseases, at different stages, and in distinct types of populations. The resting state has become the choice protocol for measuring brain connectivity. However, different brain networks are recruited under different tasks in various combinations. While looking for biomarkers in the resting state is arguably valuable and methodologically simpler (resting-state EEG is usually obtained from neurological patients as part of routine testing), it is also akin to always assessing arterial blood pressure during rest without considering that some abnormalities will only surface when the subject is performing certain types of effort. Those studies that identified qEEG biomarkers by delivering a stimulus to the subject or requiring them to complete a task effectively induced the activation of specific networks. Research on ADHD includes examples of this approach [84]. Activation will change between ill patients and healthy subjects, as evidenced by studies in cognitive neurosciences and some directly related to qEEG [85]. Most importantly, it will change in patients going through different stages of the same disease. Features like EEG microstates also reflect these changes in network recruitment and reconfiguration. Thus, experimental protocols that include tasks and activation techniques should be considered and tailored according to the type of biomarker and the disorder that the researchers wish to investigate. Biomarkers of stage and biomarkers of treatment response may benefit less from the resting-state approach than other types of biomarkers. On the other hand, the same criteria should be applied for the use of ERPs. Since their capabilities as measures depend largely on neural activity being time-locked to a stimulus presentation [86,87], brain processes with complex time-dependent features would be better assessed with other techniques.

Regarding the limitations of the present manuscript, while a state-of-the-art review is not meant to be an exhaustive treatment of the topic of interest, it should be acknowledged that the number of papers cited was kept at a moderate number deliberately. This always entails the risk of unintentionally leaving important breakthroughs out of the discussion. On the other hand, once the reader has obtained a general panorama of the field from the present work, the pursuit of their particular interests should lead them to important publications that might have been omitted here. Finally, the list of authors of this review includes neuroscientists and engineers who are all involved in biomedical research in some capacity. This might introduce a bias, particularly when discussing the value of particular publications for clinical practice, when their original intention was directed elsewhere. Nevertheless, it is hoped that this bias might be taken with a grain of salt and inform the reader about the challenges faced in the field.

## 11. Challenges and Future Directions

Despite the advancements summarized above and their potential, EEG biomarkers have several limitations and face challenges in their widespread adoption. These are related to variability in acquisition protocols, patient heterogeneity, the lack of standardized thresholds for interpretation, and the diversity of analysis methods that complicate cross-study comparisons. Also, variability in individual brain anatomy and injury characteristics necessitates further research to establish robust, universally applicable biomarkers. Ongoing studies aim to refine these techniques, enhancing their reliability and integration into routine clinical practice. Advances in machine learning and multimodal integration with imaging biomarkers may address these limitations, paving the way for personalized approaches to neuropsychiatric disease management. It should be pointed out that, as in any growing field, there is an apparent need for researchers to step back occasionally and evaluate the direction of their own work. Given the sheer volume of publications to go through, thorough systematic reviews are of pivotal importance for this process.

EEG biomarkers can also be influenced by several factors, such as sensitivity to confounding factors, where EEG signals can be affected by medications, comorbidities, and other unrelated variables. For instance, anti-seizure medications (ASMs) have been shown to alter EEG readings, particularly in the delta frequency band. A study analyzing intracranial EEG recordings during ASM tapering found a dose-dependent decrease in delta band power, indicating that medication levels can significantly influence EEG outcomes [59,86].

Finally, EEG can be affected by data integration. In this respect, combining EEG biomarkers with other modalities, such as imaging or blood biomarkers, may enhance diagnostic and prognostic accuracy in neuropsychiatric diseases. Research suggests that integrating EEG features with blood-based brain injury biomarkers can improve prognostication after pediatric cardiac arrest, implying that a multimodal approach could offer more comprehensive insights than EEG alone [87].

Future research should focus on developing standardized EEG protocols for neuropsychiatric diseases studies, exploring longitudinal changes in EEG biomarkers during recovery and integrating multimodal approaches to provide a comprehensive understanding of neuropsychiatric disease pathophysiology. As can be seen in Table 2, there is one or two types of biomarkers that are favored for each set of neuropsychiatric disease. The latter means that there is a balance to be reached between the exploration of too many approaches that might prove to be dead ends and the application of only one type of biomarker when the use of other types might benefit patients further. In the case of TBI, for instance, most work is devoted to predictive biomarkers. This is to be expected since disease progression is especially important for patients with TBI. However, the stages of modification or loss of function that these patients go through might coincide with similar stages in other disorders (like those related to cognitive impairment or ADHD). Flexibility in the application of several (but not too many) biomarkers would benefit clinical practice. Again, this can only be achieved with research that is better focused and clinicians who are adequately updated in these advancements.

## Figures and Tables

**Table 1 bioengineering-12-00295-t001:** Proposed classifications of biomarkers for neuropsychiatric diseases. Adapted from [5,6].

Reference	Proposed Biomarker Classifications
Weickert et al., 2013 [5]	*Diagnostic*: the defining characteristics shared by all patients with the disease (or most if there is more than one biomarker) *Prognostic*: the prediction of the possibility of the onset of disease *Theranostics*: prediction of response to treatment
Davis et al., 2014 [6]	*Biomarkers of risk*: the identification of at-risk individuals *Biomarkers of diagnosis/traits*: reflect the presence of a disease state and allows for a definitive diagnosis of disease, ideally with no overlap between disorders or disruption by confounders *Biomarkers of state/acuity:* reflect the present severity of a particular disease process *Biomarkers of stage:* indicate an individual’s stage of illness (per extant classifications) *Biomarkers of treatment response:* an index of the probability of response to a given treatment *Biomarkers of prognosis*: predictors of the likely course and outcome of an illness

**Table 2 bioengineering-12-00295-t002:** A summary of representative examples of the literature on qEEG with reference to a biomarker class in each case.

Disorder(s) Related to the Study	Features Analyzed	Sample Size	Main Results	Task(s) or Test(s) Used	Analysis Methods	Biomarker	Biomarker Class (Davis et al., 2014) [6]	Reference
Mild cognitive impairment	Power spectra ratios for delta, theta, alpha, beta, and gamma using single channel on frontotemporal area	10 patients with dementia; 33 patients with MCI; 77 patients with HC	Single electrode on Fp1; delta bands are increased for HC compared to dementia group and MCI group Alpha-1 bands are increased for dementia group	Resting-state EEG Eyes closed for 100 s	SDW algorithm, EEMD, STFT, SVM	Frontotemporal EEG	Diagnostic	Mitsukura et al., 2022 [7]
Tau-driven neurodegeneration	Relative gamma power	7 patients with AD with significant amyloid and tau deposition; 9 patients with either SCD or OCD without tau deposition	Gamma oscillations are increased in frontal and parietal regions (as compared to gamma oscillations in presence of amyloid)	Resting-state EEG	PSD using single Hanninf taper	Gamma power	Diagnosis/trait	Nencha et al., 2024 [8]
Subjective cognitive decline; mild cognitive decline; Alzheimer’s disease	Analyses (PSD, connectivity, and microstate markers) to identify differences between patients with SCD, MCI, and HC	57 patients with SCD; 46 patients with MCI; 19 patients with HC	In spectral analysis, three-group comparison showed differences in global field power in delta, theta, and alpha bands In connectivity analysis, results did not show difference among groups in average strengths of connection between brain areas In microstate analysis, comparison demonstrated altered topographies of microstates A and D in each clinical condition compared to controls and differences between SCD and controls for microstate C	Resting-state EEG	PSD, eLORETA, 3D-mesh model, microstates, Hurst exponent	Microstate analysis	Risk, diagnostic, stage	Lassi et al., 2023 [9]
Monitoring of early cognitive decline in Alzheimer’s disease	PSD in each frequency band (delta, slow-theta, theta, slow-alpha, alpha, slow-beta, beta, and gamma), TAR, and grand average ERP waveforms	38 patients with MCI; 44 patients with HC	In resting state with eyes closed, MCI group exhibited reduced power in slow-beta, increased power in theta, increased TAR, and no significant decline in alpha power In ERP tasks, MCI group exhibited reduced ERP and late positive potential, delayed ERP and early component latency, slower reaction time, and lower response accuracy	Resting-state EEG, 3CVT, SIR memory task	PSD, machine learning to build model that combines multiple EEG/ERP measures to obtain single unified score of MCI and SVM	Power of each EEG/ERP predictor, TAR, power in theta and beta bands, MCIsc	Diagnosis, state/acuity, treatment response	Meghdadi et al., 2024 [10]
Alzheimer’s disease	Features for delta, theta, alpha, beta, gamma, and full frequency bands using WC and RHHT in eyes closed, eyes open and eyes closed and eyes open tests	20 patients with AD; 20 patients with HC	Machine learning classification accuracy (%) shows that RHHT performs better in full band in frontocentral midline and occipital regions For WC method, best accuracy was achieved in theta band In time-frequency functional connectivity, RHHT and WC have dominating power in alpha band	Resting-state EEG	WC and RHHT, PFoCSs, CEEMDAN algorithm, PSD, machine learning	Peak frequency of cross-spectrums, estimated from RHHT	Diagnostic	Cao et al., 2022 [11]
Lewy body dementia and Alzheimer’s disease	Alpha rhythm from occipital region (peak frequency, reactivity, and power)	41 patients with Lewy body dementia (24 patients with Lewy body dementia; 17 patients with Parkinson’s disease dementia); 40 patients with HC	Marked reduction in alpha Reactivity in dementia with Lewy bodies as compared to AD and PDD	Resting-state EEG	PSD using Bartlett’s method	EEG alpha reactivity	Diagnostic (discrimination between types of dementia)	Schumacher et al., 2020 [12]
Mild cognitive impairment in Parkinson’s disease	Power average of delta, deltatheta, theta, and alpha frequency bands in frontal areas for therapeutic joint effect of interventions (CT and PA)	Patients with PD under cognitive (10 patients) or physical (9 patients) training	Positive joint effect of interventions (CT and PA) was positive on cognitive abilities and executive functions	Resting-state EEG	Power spectrum	Theta and alpha power at frontal areas	Intervention efficacy	Trenado et al., 2023 [13]
Dementia due to Parkinson’s disease	Reactivity of posterior (central, parietal, and occipital) EEG alpha rhythm	73 patients with PDD; 35 patients with ADD; 25 patients with HC	Posterior EEG alpha source activities manifested lower reactivity in parietal region	Resting-state EEG	TF, BGF, eLORETA, alpha reactivity (%)	Resting-state electroencephalographic alpha source reactivity	State/treatment response	Babiloni et al., 2024 [14]
Parkinson’s disease	Time percentage that frequency peak is within theta or alpha band; relative powers of delta, theta, alpha, beta, and gamma bands; spectral exponent; alpha/theta ratio; slope of non-oscillatory activity from average PSD	13 cognitively normal patients with PD with age-matched HC	Higher relative theta power, higher time percentage with frequency peak in theta band and reduced alpha/theta ratio, steeper non-oscillatory spectral slope activity, less alpha and beta reactivity to eyes open test	Resting-state EEG	PSD	In EO, alpha reactivity, beta reactivity, percentage of theta band, and spectral slope In EC, relative theta power and reduction in alpha/theta ratio.	Risk; diagnostic	Gimenez-Aparisi et al., 2023 [15]
Cognitive decline after mild SARS-CoV-2 infection	Features for delta, theta, alpha, beta, gamma, and full frequency using spatial, linear, and nonlinear analysis	185 patients in four age categories (<10; 10–20; 20–27; >27)	Reductions in connectivity around temporal region to frontal region; reductions in connectivity were mainly intra-hemispheric; alterations in frequency within delta and theta bands	Resting-state EEG	Source connectivity analysis using dDTF, KMeans clustering method for microstate analysis, Hjorth parameter, Kolmogorov complexity, sample entropy, Hurst index	Spatial, linear, and nonlinear biomarkers	Diagnosis/trait; state/acuity	Sun et al., 2024 [16]
Post-COVID-19 subjective cognitive decline	Frontal, central, and parietal absolute power; theta/beta power ratio; interhemispheric coherence	50 COVID-19 survivors; 50 patients with HC (aged 33–45)	Increased central and parietal theta/beta ratio; significantly lower frontal, central, and parietal coherence	Neuropsychological tests	Fast Fourier transformation Coherence (method not specified)	Coherence; theta/beta ratio	Diagnostic/trait; stage (potentially)	Gaber et al., 2024 [17]
TBI	Spectral phase and power of alpha, theta, and delta frequency bands and connectivity using debiased weighted phase lag index in order to provide diagnostic and prognostic information at 3 and 6 months post-injury	20 patients with HC	Mean relative alpha power and outcome at 3 months suggests potential for simple and standard EEG measure to augment prognostication in post-traumatic states of unresponsiveness	Resting-state EEG	Machine learning: graph theory analyses	Relative alpha power	Prognostic	O’Donnell et al., 2021 [18]
Traumatic brain injury (TBI)	RWC across EEG frequency bands within frontal, central, parietal, occipital, and temporal regions; Halstead–Reitan categorization task scores and latencies	8 patients with TBI; 8 patients with HC	Coherence values in parietal and occipital regions, in beta and gamma bands, and coherence values in temporal region in delta and theta bands	Resting-state EEG and Halstead–Reitan categorization task	EEG Spectral Power Analysis, EEG-EEG Cortico-Cortical Coherence, RWC	Higher coherence in beta and gamma frequency bands in parietal region Lower delta and theta frequency bands in temporal region	Prognostic	Méndez-Balbuena et al., 2024 [19]
Depression in healthy adults	Spatial filtering, fPCA, and conventional frequency analyses in theta band CSD and eLORETA analyses in alpha band	35 patients with HC	Weak theta band activity in resting-state EEG; posterior alpha frequency was prominent, reliably quantified, and persistent across data transformation	Resting-state EEG	CSD, fPCA, eLoreta	Alpha and theta spectral components	Diagnostic/trait	Smith et al., 2019 [20]
Prescription of antidepressants in MDD	EEG features (IED, APF, and FAA) to select among three different antidepressants (escitalopram, sertraline, or venlafaxine) as compared to TAU	122 patients: 70 with EEG informed prescription; 52 patients undergoing treatment as usual	EEG-informed prescription group demonstrated significantly better response relative to TAU group Sertraline was used in abnormal EEG activity (IEDs, slowing of the EEG, and APF below 8 Hz) Escitalopram or sertraline was advised for right-sided FAA Venlafaxine was advised for left-sided FAA	Resting-state EEG	Decision tree algorithm	IEDs or slowing of EEG, AFP below 8 Hz, FAA	Treatment response	van der Vinne et al., 2020 [21]
Depression recognition	Linear features, nonlinear features, and functional connectivity features (PLI) in EEG signals	24 diagnosed patients, 29 patients with HC	Highest accuracy (82.31%) in linear, nonlinear, and PLI using LR classifier and method ReliefF, set Number of functional connectivity features (PLI) is higher than number of nonlinear and linear features PLI and nonlinear features showed significant differences Connection edges of left hemisphere > connection edges of left hemisphere Intrahemispheric connection edges > inter hemispheric connection edges	Resting-state EEG	Machine learning: CFS, Information Gain, and ReliefF LR, KNN, DT, and NB classifiers	Intrahemispheric connection edges of PLI	Diagnostic	Sun et al., 2020 [22]
Epilepsy in childhood	Rates of ripples, spikes ripples, and spikes per minute	26 patients with epilepsy; 30 patients with HC	Higher rates of ripples and spike ripples in early EEG Ripple rate was significantly higher in seizure group than in no-seizure group	EEG awake and/or asleep	Machine learning: ROC curves, AUC, Youden index	High frequency oscillations	Risk, diagnostic/trait	Klotz et al., 2020 [23]
Migraine	Resting-state alpha band oscillations in visual area of brain	13 patients with migraine; 17 patients with HC	Lower alpha band (8 to 10 Hz) augmented power	Resting-state EEG before and after contrast detection task	Spectral analysis	Increase in lower alpha band power	Basic research; has potential as biomarker of risk	O’Hare et al., 2018 [24]
Epilepsy and migraine	Classification accuracy and effect of number of neurons in BilLSTM, RSNN, and NeuCube models from Fp1, Fp2, C3, C4, O1, O2, F7, F8, T3, T6, and Cz EEG channels	6 patients with epilepsy; 15 patients with migraine; 15 patients with HC; both sexes, aged 6 to 57	BiLSTM identifies F8, T3, and T6 as crucial EEG channels on classification, while RSNN highlights F7 and T6, and NeuCube suggests C4, F8, T6, and F7 as discriminative channels BiLSTM models and reveals asymmetry between high and low activity in some channels, particularly in occipital lobe	Resting-state EEG	Machine learning methods: deep BilLSTM classifier, RSNN, and NeuCube	Stronger EEG channel activities across models, especially F8 and C4, contribute to understanding of epilepsy and migraine disorders; spike generation and spike exchange in NeuroCube model	Diagnosis (discrimination between epilepsy and migraine; prediction of crises)	Saeedinia et al., 2024 [25]

AD, Alzheimer’s disease; ADD, Alzheimer’s disease dementia; APF, alpha peak frequency; AUC, area under the curve; BGF, background frequency; BilLSTM, bi long short-term memory; CEEMDAN, complete ensemble empirical mode decomposition with adaptive noise; CFS, correlation feature selection; CSD, current source density; CT, cognitive training; dDTF, direct directed transfer function; DT, decision tree; EEMD, ensemble empirical mode decomposition; eLORETA, exact low-resolution brain electromagnetic tomography; EO, eyes open; EC, eyes closed; ERP, event-related potential; FAA, frontal alpha asymmetry; FESZ, first-episode schizophrenia; fPCA, functional principal component analysis; HC, healthy control; HR, high risk; IED, interictal epileptiform discharges; KNN, k-nearest neighbor; LR, logistic regression; LSTM, long short-term memory; MCI, mild cognitive impairment; MCIsc, mild cognitive impairment composite score; NB, naïve bayes; OCD, objective cognitive deficits; PA, physical activity; PD, Parkinson’s disease; PDD, Parkinson’s disease dementia; PfoCSs, peak frequency of cross-spectrums; PLI, phase lag index; PSD, power spectral density; RHHT, revised Hilbert–Huang transformation; ROC, receiver operating characteristic; RSNN, reservoir spiking neural network; RWC, regional weighted coherence; SCD, subject cognitive decline; SDW, summation of derivatives within windows; SIR, standard imagen recognition; STFT, short-time Fourier transform; SVM, support vector machines; TAR, theta-to-alpha ratio; TAU, treatment as usual; TF, transition frequency; UHR, ultra-high-risk; WC, wavelet cross-spectrum; 3CVT, 3-choice vigilance task.

## Data Availability

No new data were created or analyzed in this study.

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
