# Peer review of "Electroencephalographic Biomarkers for Neuropsychiatric Diseases: The State of the Art"

_bioengineering, 2025, doi:10.3390/bioengineering12030295_

Round 1

Reviewer 1 Report

Comments and Suggestions for Authors

This paper “Biomarkers for Cortical Neuropsychiatric Diseases: A Primer” by Huidobro et al. is a review article on the diagnostic use of EEG for various neuronal diseases. There are several typos and errors in the citations that need to be corrected before finalizing for publications as listed below:

L32:      I believe this meant to be the first citation. There is no [1] in this manuscript.

L133:   “serious cognitive decline of dementia” requires citation.

L224:   “populati” should be population.

L260:   (Zheng et al. 2023) is [29]?

Reviewer 2 Report

Comments and Suggestions for Authors

Dear Editor,
I am ready to submit my review on the manuscript titled "Biomarkers for cortical neuropsychiatric diseases: a primer", which has been submitted to BS. 

The study aims to provide a foundational understanding of biomarkers, particularly those related to EEG, in the context of cortical neuropsychiatric diseases based on selected literature. It seeks to define what biomarkers are, discuss their various types and applications, and highlight their potential to improve diagnosis, treatment, and research for these complex conditions. Among the conditions of interest are cognitive impairment, traumatic brain injury, depression, migraine, and epilepsy. The study concludes that biomarkers hold significant promise for advancing our understanding and treatment of cortical neuropsychiatric diseases.

Among the strengths of the study are the interesting topic and the well-written manuscript. It is clear that the authors did a commendable job. However, there are several notable weaknesses. First, the lack of a systematic and reproducible literature search raises a significant risk of selection bias. Second, the authors did not assess the quality of the eligible studies, which further increases the overall risk of bias. Third, they failed to discuss the limitations of their own study. Finally, the conclusions drawn are vague and overly general.

Correcting the limitations mentioned above is expected to significantly alter the manuscript draft. Therefore, I recommend against publishing the current version.However, the review has several significant limitations.

Best regards

Reviewer 3 Report

Comments and Suggestions for Authors

This submission is of a review paper on ‘biomarkers’ for ‘cortical neuropsychiatric’ disorders. The authors searched the literature for electroencephalographic (ie EEG) biomarkers for neuropsychiatric diseases of cortical origin. They summarised the information of various EEG patterns seen in these disorders. This paper can be a helpful summary for those new to this field.

However, there are some issues the authors may wish to address.

The most major is the way the paper is organised, the information provided (or not), the text vs the tables. Each disease should be listed in a clinically logical and non-repetitive manner (eg ‘TBI’ and migraine and Parkinsons’ disease appear twice, depression a few times, all separately in the un-numbered un-titled table). For each disease being covered, the information should be similarly structured eg definition, epidemiology, pathophysiology, EEG findings and which aspect that information covers as per Weickert or Davis(the authors seem to prefer this, which to me is a better, more granular and clinically useful classification) (Table 1). The text and tables (especially the currently un-numbered and untitled table) should be in sync.

More specifically:

1.      Tile – suggest adding ‘Electroencephalographic’ as the first word, to reflect that this is the biomarker being reviewed

2.      (major) Abstract – to summarise the actual key findings rather than merely mention what the review will focus on. Remember that, for the reader, the abstract is the first view of the paper after the title and should have enough information to briefly educate the reader (who may be looking through many abstracts) on the issue, and hopefully to draw her/him to read the full paper

3.      Line 45 - please mention examples of ‘neuropsychiatric diseases’ that will be covered later in the paper

4.      (major) Insert paragraph on what are ‘cortical’ neuropsychiatric diseases. Are there ‘subcortical’, ‘deep’, ‘brainstem’ or ‘cerebellar’ neuropsychiatric diseases too, that warrant the term ‘cortical’? Lines 45-51 are a good start

5.      (major) Insert a paragraph briefly explaining the various EEG terms used in the paper. Lines 139-146 are a good start

6.      Lines 166 and 167 – isn’t ‘MCI’ a type of ‘cognitive decline’?

7.      Line 224 – seems incomplete

8.      2nd table - no table number, title

9.      Lines 253-257 – should ADHD be a separate section?

10.  Line 270 - ‘qEEG Biomarkers for’ can be removed to be consistent with the other headings

11.  (major) Lines 499-505, 509-521 – best under a separate section ‘Limitations of the Technique’, and expanded. Then have a new Section ‘Potential Solutions’

12.  (major) Suggest adding a new table that is similar to the untitled table in column 1, then the subsequent columns are 1 each of the Davis classification, and the boxes can be filled with the available information. This will show where the knowledge gaps are and should spur the necessary research to fill these gaps

13.  The following references may be helpful:

1: Parsa M, Rad HY, Vaezi H, Hossein-Zadeh GA, Setarehdan SK, Rostami R, Rostami

H, Vahabie AH. EEG-based classification of individuals with neuropsychiatric

disorders using deep neural networks: A systematic review of current status and

future directions. Comput Methods Programs Biomed. 2023 Oct;240:107683. doi:

10.1016/j.cmpb.2023.107683. Epub 2023 Jun 20. PMID: 37406421.

2: Bogéa Ribeiro L, da Silva Filho M. Systematic Review on EEG Analysis to

Diagnose and Treat Autism by Evaluating Functional Connectivity and Spectral

Power. Neuropsychiatr Dis Treat. 2023 Feb 22;19:415-424. doi:

10.2147/NDT.S394363. PMID: 36861010; PMCID: PMC9968781.

3: Michelini G, Salmastyan G, Vera JD, Lenartowicz A. Event-related brain

oscillations in attention-deficit/hyperactivity disorder (ADHD): A systematic

review and meta-analysis. Int J Psychophysiol. 2022 Apr;174:29-42. doi:

10.1016/j.ijpsycho.2022.01.014. Epub 2022 Feb 4. PMID: 35124111.

4: Shirahige L, Berenguer-Rocha M, Mendonça S, Rocha S, Rodrigues MC, Monte-

Silva K. Quantitative Electroencephalography Characteristics for Parkinson's

Disease: A Systematic Review. J Parkinsons Dis. 2020;10(2):455-470. doi:

10.3233/JPD-191840. PMID: 32065804; PMCID: PMC7242841.

5: Helm K, Viol K, Weiger TM, Tass PA, Grefkes C, Del Monte D, Schiepek G.

Neuronal connectivity in major depressive disorder: a systematic review.

Neuropsychiatr Dis Treat. 2018 Oct 17;14:2715-2737. doi: 10.2147/NDT.S170989.

PMID: 30425491; PMCID: PMC6200438.

Round 2

Reviewer 2 Report

Comments and Suggestions for Authors

The authors addressed all concerns adequately.

Author Response

Dear reviewer, thank you again for your comments and suggestions.

Reviewer 3 Report

Comments and Suggestions for Authors

This is a revised submission of a review paper on electroencephalographic biomarkers for neuropsychiatric’ diseases. I thank the authors, the paper is much improved.

However, there are some mostly minor issues the authors may wish to address:

  1. Abstract – where is the change?
  2. Line 21 – is ‘of cortical origin’ still necessary?
  3. Line 22 - please spell out ‘qEEG’ in full at first use
  4. Line 23 - please spell out ‘TBI’ in full at first use
  5. Line 27 – ‘arte’ should be ‘art’
  6. Lines 131-138 – I suggest a new section titled ‘Search Strategy’
  7. Line 134 – to spell out ADHD in full at first use; ‘traumatic brain injury’ has been spelled out in full earlier and can be removed
  8. Line 252 – to add ‘(SCI)’ after ‘impairment’
  9. Line 264 – suggest adding ‘EEG’ after ‘State’
  10. Line 366 – is this section about EEG during tasks? Then please state so, so that it contrasts with ‘resting state EEG’
  11. Table 2 – I suggest that the data for ref 8 be moved up to before ref 19
  12. (major) – I do not see any section on schizophrenia although it is in the table 2
  13. References – 12, 24.15 – missing some authors’ names…

Author Response

Dear reviewer,

We thank you for your valuable observations and comments. All of the abbreviation errors and typos you pointed out have been corrected in their respective line in the manuscript. Regarding particular points:

  1. Abstract – where is the change?

The abstract was changed per your suggestion in the past revision of the manuscript. Starting with “Our findings suggest that…” on line 23, we include a brief summary of the observations derived from our state-of-the-art review.

  1. Lines 131-138 – I suggest a new section titled ‘Search Strategy’

We appreciate your suggestion. However, we believe this paragraph is too brief to require a separate section.

  1. Line 366 – is this section about EEG during tasks? Then please state so, so that it contrasts with ‘resting state EEG’

Indeed, this section deals with task oriented qEEG for cognitive and motor processing. The heading in line 366 has been changed accordingly.

  1. Table 2 – I suggest that the data for ref 8 be moved up to before ref 19

This reference has been moved per your suggestion. Please note also that the appropriate reference numbers have been added to the citation column in table 2, as the numbers in the first column (now removed) actually indicated row numbers and were a remnant of the corrections we made to the table.

  1. (major) – I do not see any section on schizophrenia although it is in the table 2

We have removed this reference in table 2, since it was not reflected in the text.

  1. References – 12, 24.15 – missing some authors’ names…

These and other references have been re-formatted according to the journal’s guidelines.